# Proteomic and Bioinformatic Tools to Identify Potential Hub Proteins in the Audiogenic Seizure-Prone Hamster GASH/Sal

**DOI:** 10.3390/diagnostics13061048

**Published:** 2023-03-09

**Authors:** Carlos García-Peral, Martín M. Ledesma, M. Javier Herrero-Turrión, Ricardo Gómez-Nieto, Orlando Castellano, Dolores E. López

**Affiliations:** 1Instituto de Neurociencias de Castilla y León (INCYL), Universidad de Salamanca, 37007 Salamanca, Spain; 2Instituto de Investigación Biomédica de Salamanca (IBSAL), 37007 Salamanca, Spain; 3Unidad de Conocimiento Traslacional, Hospital de Alta Complejidad del Bicentenario Esteban Echeverría, Monte Grande B1842, Argentina; 4Instituto de Fisiopatología y Bioquímica Clínica (INFIBIOC), Hospital de Clínicas “José de San Martín”, Facultad de Farmacia y Bioquímica, UBA, Buenos Aires C1053, Argentina; 5Banco de Tejidos Neurológicos del INCYL (BTN-INCYL), 37007 Salamanca, Spain; 6Department de Biología Celular y Patología, Facultad de Medicina, Universidad de Salamanca, 37007 Salamanca, Spain

**Keywords:** epileptogenesis, epileptogenic nucleus, GASH/Sal hamster strain, genetic epilepsy models, proteomic profile

## Abstract

The GASH/Sal (Genetic Audiogenic Seizure Hamster, Salamanca) is a model of audiogenic seizures with the epileptogenic focus localized in the inferior colliculus (IC). The sound-induced seizures exhibit a short latency (7–9 s), which implies innate protein disturbances in the IC as a basis for seizure susceptibility and generation. Here, we aim to study the protein profile in the GASH/Sal IC in comparison to controls. Protein samples from the IC were processed for enzymatic digestion and then analyzed by mass spectrometry in Data-Independent Acquisition mode. After identifying the proteins using the UniProt database, we selected those with differential expression and performed ontological analyses, as well as gene-protein interaction studies using bioinformatics tools. We identified 5254 proteins; among them, 184 were differentially expressed proteins (DEPs), with 126 upregulated and 58 downregulated proteins, and 10 of the DEPs directly related to epilepsy. Moreover, 12 and 7 proteins were uniquely found in the GASH/Sal or the control. The results indicated a protein profile alteration in the epileptogenic nucleus that might underlie the inborn occurring audiogenic seizures in the GASH/Sal model. In summary, this study supports the use of bioinformatics methods in proteomics to delve into the relationship between molecular-level protein mechanisms and the pathobiology of rodent models of audiogenic seizures.

## 1. Introduction

Epilepsy is a chronic neurological disorder characterized by abnormal neuronal activity arising from imbalances between excitatory and inhibitory synapses [1,2], which are highly correlated with functional and structural changes in specific regions of the brain [3,4]. The difference between the normal and epileptic brain may harbor genetic variations, changes in gene expression, and/or protein alterations in the epileptogenic nucleus [5,6,7]. It is increasingly clear that such differences contribute to the development of different epilepsy phenotypes.

The main challenges in epilepsy research include understanding disease progression and elucidating the various manifestations of epilepsy by searching for new molecular biomarkers [8]. In this sense, the application of molecular techniques to carry out exhaustive studies on nucleic acids (DNA and RNA) and protein levels is extremely important to elucidate molecular dysregulations in the epileptic brain.

Proteomics research can offer a comprehensive understanding of the molecular processes that underpin different biological states, such as healthy or diseased processes. The literature establishes that the association of proteins with epilepsy is mainly related to synaptic physiology, energy metabolism, and in the cellular cytoskeleton and metabolic enzymes [9,10,11]. There are also numerous studies that describe the role of mitochondrial proteins in the epilepsy etiology and in the refractory epilepsy [12,13], including synaptic physiology, cell structure, cell stress, metabolism, and energetics.

Within all of these reports, proteomics studies are more focused on the changes that occur after seizure induction in experimental animals [14,15,16,17], in patients after years of seizures and epilepsy [9,18,19], or specifically to determine the changes produced after treatment with antiepileptic drugs [11,20].

However, there are few studies that describe the protein modifications in the epileptogenic nucleus that trigger the seizures before they occur; thus, protein disruptions capable of generating seizures require further study. In epilepsy research, experimental animal models are essential to understand the mechanisms underlying the genesis, establishment, and progression of seizures and epilepsy [21,22]. Among these models, the strains of animals with genetic audiogenic susceptibility for developing seizures in response to high-intensity acoustic stimulation [23,24] can be used to carry out protein difference studies before, during, and after seizures [25].

The Genetic Audiogenic Seizure Hamster (GASH/Sal), a hamster strain inbred at the University of Salamanca, exhibits genetic audiogenic epilepsy similar to human tonic–clonic seizures [26], shows an autosomal recessive inheritance for susceptibility to audiogenic seizures [27], and presents genetic differences in comparison with the wild-type [28]. It has already been validated as a model of epilepsy in fundamental aspects at the behavioral [27], electroencephalographic [26], pharmacological [29,30], molecular [28,31,32], and morphological [33,34] levels.

The behavioral manifestations of the seizures in the GASH/Sal model are initiated very quickly (7–9 s) after high-intensity acoustic stimulation [27]. As occurred in other audiogenic seizure models, the inferior colliculus (IC), a critical integration center in the auditory midbrain pathway, receives altered bottom-up auditory inputs that generate the seizure-prone network [34]. This implies the existence of innate protein disturbances in the IC, as a basis for seizure susceptibility and generation in the GASH/Sal model. Further study of the proteomic profile in the epileptogenic nucleus is critical, as it would provide valuable information on the molecular mechanisms underlying the regulation of neuronal excitability, which could shed light on the specific factors that contribute to seizure susceptibility in this strain.

## 2. Materials and Methods

### 2.1. Experimental Animals

A total of 12 golden hamsters, *Mesocricetus auratus*, 6 GASH/Sal, from the animal facility of the University of Salamanca, and 6 control hamsters (control group) from Janvier Labs (Le Genest-Saint-Isle, France) were used in this study. All were males between 2 and 4 months of age. We selected animals of that age because in this period the GASH/Sal exhibits the maximum susceptibility to seizures [27]. All the GASH/Sal were naïve, without receiving any acoustic stimulation to trigger audiogenic seizures. All procedures involving animals and their care were conducted in accordance with the guidelines for the use and care of laboratory animals of the European Communities Council Directive (2010/63/EU), the current Spanish legislation (RD 1201/05), and with those established by the Institutional Bioethics Committee (approval number 375).

### 2.2. Protein Extraction and Purification

The IC of each animal was obtained following euthanasia by deep anesthetization (pentothal overdose) and rapid decapitation. Samples were ground into powder in liquid nitrogen and then extracted with lysis buffer (7 M urea, 2 M thiourea, 4% CHAPS, 40 mM TRIS hydrochloride (tris-HCl), pH 8.5) containing 1 mM PMSF and 2 mM EDTA. Next, 10 mM DTT was added, and after 5 min, the suspension was sonicated at 200 W for 15 min and then centrifuged at 4 °C and 25,000× *g* for 15 min. The supernatant was mixed well with a 5× volume of chilled acetone containing 10% (*v*/*v*) TCA and incubated at −20 °C for 30 min. Following centrifugation at 4 °C and 25,000× *g* for 15 min, the supernatant was discarded. Then, the pellet was air-dried and dissolved in lysis buffer (7 M urea, 2 M thiourea, 4% NP-40^®^, 20 mM tris-HCl, pH 8.0–8.5). The suspension was sonicated at 200 W for 15 min and centrifuged at 4 °C and 25,000× *g* for 15 min; the supernatant was the protein solution. Proteins were quantified using the Bradford method [35], separated by SDS-12% polyacrylamide gels, and visualized by Coomassie blue staining. Subsequently, 100 μg of protein solution was hydrolyzed with 2.5 μg of trypsin enzyme (Trypsin, Mass Spec Grade) with the ratio of protein:trypsin of 40:1 (*w*/*w*) at 37 °C for 4 h, and enzymatic peptides were desalted using a Strata X SPE desalting column (Strata X 33 μm polymeric reversed-phase column; Phenomenex, Torrance, CA, USA). Disulfide bonds were reduced with DTT (DL-Dithiothreitol) with the final concentration of 10 mM, and then placed in a water bath at 56 °C for 1 h. Cysteines were alkylated with IAM (iodoacetamide) at a final concentration of 55 mM, placed in a dark room for 45 min, and vacuumed to dryness.

### 2.3. High-pH Reversed-Phase (RP) Separation

Equal amounts of peptides were extracted from all samples and mixed, and the mixture was diluted with mobile phase A (5% can, pH 9.8) and injected. The Shimadzu LC-20AB HPLC system coupled with a Gemini high pH C18 column (5 μm, 4.6 × 250 mm) was used. The sample was subjected to the column and then eluted at a flow rate of 1 mL/min by gradient: 5% mobile phase B (95% ACN, pH 9.8) for 10 min, 5% to 35% mobile phase B for 40 min, 35% to 95% mobile phase B for 1 min, flow Phase B for 3 min, and 5% mobile phase B equilibrated for 10 min. The elution peak was monitored at a wavelength of 214 nm, and the component was collected every minute. Components were combined into a total of 10 fractions, which were then freeze-dried. The ten high pH fractions were subjected to Data-Dependent Acquisition (DDA), and the unfractionated to DIA.

### 2.4. DDA and DIA Analysis by Nano-LC-MS/MS

The dried peptide samples were reconstituted with mobile phase A (2% ACN, 0.1% FA) and centrifuged at 20,000× *g* for 10 min, and the supernatant was taken for injection. Separation was carried out by a Thermo UltiMate 3000 UHPLC liquid chromatograph. The sample was first enriched in the trap column and desalted, then entered a tandem self-packed C18 column (150 μm internal diameter, 1.8 μm column size, 35 cm column length), and was separated at a flow rate of 500 nL/min by the following effective gradient: 0~5 min, 5% mobile phase B (98% ACN, 0.1% FA); 5~120 min, mobile phase B linearly increased from 5% to 25%; 120~160 min, mobile phase B increased from 25% to 35%; 160~170 min, mobile phase B increased from 35% to 80%; 170~175 min, 80% mobile phase B; 175~180 min, 5% mobile phase B. The nanoliter liquid phase separation end was directly connected to the MS.

On one side, each of the ten high pH fractions (fractioned peptide sample) was subjected to DDA analysis to construct the spectral library. A spectral library collects all detectable non-redundant, high-quality peptide information (MS/MS spectra) of the sample, which can be used as a peptide identification template for subsequent data analysis. It contains the fragment ion intensity and retention time that characterize the peptide spectrum.

For DDA analysis, LC-separated peptides were ionized by nanoESI and injected into a tandem mass spectrometer (MS) Q-Exactive HF X (Thermo Fisher Scientific, San Jose, CA, USA) with DDA mode. The main settings were: ion source voltage 1.9 kV; MS scan range 350~1500 *m*/*z*; MS resolution 120,000, maximal injection time (MIT) 100 ms; MS/MS collision type HCD, collision energy NCE 28; MS/MS resolution 30,000, MIT 100 ms, dynamic exclusion duration 30 s. The start *m*/*z* for MS/MS was fixed to 100. The precursor for the MS/MS scan satisfied the following: charge range 2+ to 6+, top 20 precursors with intensity over 5E4. AGC was MS 3E6 and MS/MS 1E5.

Conversely, an unfractionated peptide sample from an individual sample was subjected to data-independent acquisition (DIA, also called SWATH) mode, which utilizes the latest high-resolution MS to acquire peptide ion characteristics in mass and retention time-space simultaneously. Compared to the traditional technique of extracting a single ion for fragmentation analysis, the MS is set to a wide precursor ion window in DIA mode to collect product ions. Thus, the complete collection of all detectable protein peak information in the sample and high-reproducible analysis of many samples is achieved.

For DIA analysis, LC-separated peptides were ionized by nanoESI and injected into a tandem mass spectrometer Q-Exactive HF X with DIA (data-independent acquisition) detection mode as per the following settings: ion source voltage 1.9 kV; MS scan range 400~1250 *m*/*z*; MS resolution 120,000; MIT 50 ms; 400~1250 *m*/*z* was equally divided into 45 continuous windows in the MS/MS scan. MS/MS collision type HCD, MIT (maximal injection time) was the auto mode. Fragment ions were scanned in Orbitrap, MS/MS resolution 30,000; collision energy was the distributed mode: 22.5, 25, 27.5; AGC was 1E6.

The experimental workflow is shown in Figure 1.

### 2.5. Data Analysis for Protein Identification

The protein identification analysis was made by the BGI company (Shenzhen, China). Identification and quantification of peptides and proteins were obtained from DDA spectral library by deconvolution of the DIA data.

The spectral library required by DIA analysis was constructed by merging the DDA search results from the samples that were treated with pooling and fractionation and the available database. Briefly, the Andromeda search engine within MaxQuant identified DDA data, and identification results were used for spectral library construction. The mProphet [36] algorithm was used for large-scale DIA data to complete analytical quality control, thus obtaining many reliable quantitative results. The DIA data was analyzed using the iRT peptides for retention time calibration.

Then, one percent false discovery rate (1% FDR) using a Target-Decoy database search was used for peptide validation, and only protein groups with at least one unique peptide identified were considered.

### 2.6. Database Selection

The selection of database is an important step in MS-based protein identification; the final identified protein sequences are from the selected database. Currently, databases in use can be divided into two main categories:(1)UniProt protein database

UniProt is the most informative and resourceful protein database (http://www.uniprot.org/, accessed on 14 December 2021). It consists of data from three major databases, i.e., Swiss-Prot, TrEMBL, and PIR-PSD. It is a data set verified by experts and consists of two parts: UniProtKB/Swiss-Prot (with reviewed, manually annotated entries) and UniProtKB/TrEMBL (with unreviewed, automatically annotated entries). In general, it is recommended to give priority to the subset of UniProtKB/Swiss-Prot for protein identification. When it aims to find novel sequences (such as alternative splicing or new transcripts) or to identify allied species, the UniProtKB/TrEMBL database can be considered [37].

(2)The protein databases based on genome annotation

The databases mainly include a series of databases derived from NCBI and Ensembl gene annotation databases.

Among them, we selected the protein database from the reference sequence (RefSeq) of NCBI (https://www.ncbi.nlm.nih.gov/refseq/, accessed on 14 December 2021), which is a non-redundant proteome database. It is widely used in the analysis of multi-omics studies due to the importance of the NCBI annotation system. NCBI’s RefSeq provides reference sequence for molecules that are naturally involved in central dogma, from chromosomes to mRNA and proteins. The RefSeq standard provides a basis for functional annotation of the human genome. It provides a stable reference for mutation analysis, gene expression studies, and polymorphic discovery. In addition, NCBI provides a completed non-redundant protein sequence database (NCBI_nr), including animals, plants, microbes, bacteria, and other taxonomy. Since the database is derived from various sources (including GenBank, RefSeq, SwissProt, PDB, etc.), unless the species is without complete genome annotation, or it is necessary to search for homologous data, it is not recommended to use this database for protein identification.

Ensembl aims to develop a software package with automatic annotation and maintenance for the eukaryotic genome. Ensembl has relatively complete and consistent genomic, transcriptome, and proteomic annotation information, which is ideal for multi-omics analysis.

### 2.7. Bioinformatic Analysis

This process was based on the sample data generated from a high-resolution MS.

Peptides and proteins were quantified using the MSstats software package [38], which included intra-system error correction and normalization for each sample. The database containing the log abundance values for each protein in each sample was then analyzed with the RStudio software. The analysis pipeline included the following steps: first, a filter for log abundance values missingness (NA) was applied, with a requirement of at least four actual replicate values within each group and a total of three allowed NA values per protein. After the NA cleaning, the dataset underwent sample mean imputation for the remaining NAs. The dataset was then subsetted by each replicate, with any remaining NA values replaced by the sample mean. Next, a *t*-test was mapped robustly within each protein and between groups using the purrr R package, with a cut-off value of 0.05. The dataset was then split into groups, and the mean and standard deviation (SD) of the abundance values were calculated for each protein within each group. The mean abundance differences (fold change, FC) were computed for each protein, along with the pooled SD, which is the square root of the sum of square SD (SD control hamsters and SD GASH/Sal) divided by 2. The Cohen’s effect size [39], which is the FC divided by the pooled SD, was computed as a statistical measurement of the effect size, with a cut-off value of 2 for up-regulated proteins and −2 for down-regulated proteins.

Group-exclusive proteins were assigned to those entries that fulfill the criteria of having all (six) missing values in one group (control or GASH/Sal) and simultaneously presenting at least three actual abundance values in the other group.

Clustering methods are used to classify data samples based on their similarity to other objects, and this process relies on methods for measuring the distance or (dis)similarity between observations. In this work, we computed the Euclidean distance between the rows of the data matrix and the Hierarchical Clustering to cluster all proteins and the differential proteins at the sample level. The former analysis was represented in the so-called Clustered HeatMap. Principal component analysis (PCA) is a method for dimensionality reduction that combines multiple variables into a new set of integrated ones and then selects several to represent as much original information as possible, thus achieving dimension reduction. PCA is mainly used to observe the trend of separation between groups in the experimental model and whether there are unique value points that reflect the inter- and intra-group variations from the original data. The analysis of PCA was performed after the computation of the *k*-means clustering (Euclidean was the distance, and k was set to 2).

A Monte-Carlo simulated Chi-squared test was computed to assess the relationship between the groups (Control and GASH/Sal) and the variables KOG functions, GO Molecular functions, GO Cellular component, and GO Biological process. Furthermore, a posterior analysis of Pearson residuals was conducted to determine which factors within the former variables are related to each group. The association between factors was significant if Pearson residuals were above 2 or below −2.

#### 2.7.1. Non-Bibliographic Based Approaches to Identify Hub DEPs

-Outliers

To determine outliers’ presence within the DEP log FC (LFC) values distribution, we first computed the modulus of LFC; then, the mean and SD were calculated. An outlier parametric cut-off value was implemented as if the point LFC value was higher than the mean plus two times the SD.

-Protein–Protein Interaction (PPI) network analysis

Human homolog proteins in DEPs were extracted to perform Protein–Protein Interaction (PPI) network analysis. This biological PPI database collects and rates evidence from many sources, such as experimental data, computational forecasting methods, and public text collections, providing a comprehensive picture of all known and predicted associations between proteins, including functional and physical interactions [40]. All PPI pairs with the highest confidence interaction score (>0.9) were selected. Additionally, Cytoscape 3.9.0 plugin *cytoHubba* was implemented to search for important nodes in the PPI network through three different topological analyses. Cytoscape is an open software project for integrating biomolecular interaction networks with high-throughput expression data into a unified conceptual framework [41,42]. *cytoHubba*, a Cytoscape plugin designed by Chin et al., allows users to rank and explore nodes in a network by their biological features via 11 different topological analyses [43]. Those proteins with higher degree values are more likely to be essential within the biological network. All *cytoHubba* methods can be divided into two major categories: local and global methods. Local methods are better for identifying essential proteins in networks, because they focus solely on the connections between nodes and their immediate neighbor proteins [43]. To achieve the goals of our study, we utilized three locally based methods, namely, the maximum neighborhood component (MNC), maximal clique centrality (MCC), and Degree. Hub proteins with a score ≥ 2 were identified by each topological analysis, and overlapping ones in the three methods were chosen for further discussion.

#### 2.7.2. Bibliographic Based Approach to Identify Hub DEPs

-Gene-disease association analysis

We applied the Disease 2.0 database to conduct the bibliography search of epilepsy-related genes among 178 human homolog DEPs. This bioinformatics database is a comprehensive and regularly updated resource for disease–gene associations. It systematically integrates and assigns confidence scores to evidence from various sources, providing a reliable and detailed overview of the associations between diseases and genes. Scientific evidence is divided into three channels, depending on source and methodology: (i) the knowledge channel, which comprises three curated databases (AmyCo, MedlinePlus, and UniProtKB); (ii) the experimental channel, which extracts genome-wide association studies (GWAS) information from the Target Illumination by GWAS Analytics (TIGA) database; and (iii) the text mining channel, focused on an automatic text mining of biomedical literature [44]. All three mentioned channels were consulted, and gene epilepsy associations with a confidence score > 2 were screened for discussion.

## 3. Results

Data-independent acquisition mode used in the MS quantified a total of 43,325 peptides and 5775 proteins. After following the established procedure, the first filter for missingness of log abundance values (NA) was applied, which led to the exclusion of 521 proteins from the analysis. Thus, a total of 5254 proteins were included in the subsequent analysis. Within the GASH/Sal group, the NA cases ranged between 59 and 108, and for the control group, between 61 and 105. Table 1 shows an overview of the quantitative results for each of the 12 samples.

### 3.1. Principal Component Analysis (PCA) and Clustering with All Proteins

As a quality control method, we carried out a principal component analysis (PCA). The *X*-axis is the first principal component, and the *Y*-axis is the second principal component. The first component accounts for 16% of the total variation in the data, while the second component accounts for 14%. Both summed up to 30% of the total variation (Figure 2A), which is low considering the consensus 70% variation for significance. Individual PCA shows that the 95% confidence ellipses are superimposed, and replicates 2 and 5 of the control group fall within the GASH/Sal cluster. Moreover, the Clustered HeatMap appears as a single color, meaning replicates do not cluster in the respective groups (Figure 2B). Altogether, the overall expression of the detected proteins did not allow for differentiation between the two groups.

### 3.2. Group-Exclusive Proteins

In the IC proteomic analysis, seven proteins were uniquely assigned in the control (Table 2), and 12 proteins were uniquely found in the GASH/Sal (Table 3).

### 3.3. Differentially Expressed Proteins (DEPs)

A total of 623 proteins had a *t*-test *p*-value less than 0.05. Among them, 613 proteins belonged to non-imputed proteins, while six, three, and one proteins achieved one, two, and three imputations, respectively. Of these 623 significant proteins, 184 achieved a Cohen’s effect size value higher than 2 (126 up-regulated) or lower than −2 (58 down-regulated) (see Appendix A). The 126 proteins overexpressed in the IC of the GASH/Sal animals ranged from 2 to 6.7 FC, with an increase average of 2.66 FC, and the 58 downregulated ranged from −2 to −11.04 FC, with a decrease average of −3.01 FC (Figure 3).

An interactive volcano plot is shown in Appendix A, presenting the filtered DEPs visually.

#### 3.3.1. DEPs: GO Analysis and KOG Functions

We carried out a GO function annotation analysis on the 184 DEPs (*p* < 0.05) that achieved a value higher than 2 (126 up-regulated) or lower than −2 (58 down-regulated). For the GO entries involved in the three ontologies (cellular component, biological process, and molecular function), a statistical chart was created (Figure 4A–C). We also compared the DEPs with the KOG database to predict their possible functions and perform functional classification (Figure 4D).

Upon analyzing the entries that exhibit significant differences between the groups, we observed that the overexpressed DEPs are predominantly located in the nucleus and are mainly involved in metabolism affecting RNA processing and modification as well as translation processes. On the other hand, underexpressed DEPs are involved in a greater number of molecular functions and biological processes than the overexpressed DEPs. This suggested that underexpressed DEPs affect a greater number of cellular functions.

The list of proteins that show significant differences between the groups in each entry are shown in Appendix A.

#### 3.3.2. Protein–Protein Interaction (PPI) Network Analysis

In principle, 186 DEPs, excluding those without human counterparts, were evaluated in the STRING database (https://string-db.org/, accessed on 17 October 2022). However, one of the DEPs, TSTD3 (underexpressed −4.434 FC) is not recognized by STRING; therefore the network is built with 178 human homolog proteins.

A PPI network with an interaction score > 0.9 was constructed in the Cytoscape platform. The interaction enrichment analysis yielded a significant result (*p*-value = 0.000372), indicating that there are more interactions among the proteins found in the analyses than would be expected in a random set of proteins.

Our interaction network consists of 53 nodes (proteins) and 44 edges (junctions or interactions). The essential proteins in this network are highlighted in yellow in Figure 5.

Subsequently, the *cytoHubba* plugin was applied to determine nodes with a SCORE ≥ 2 in three topological analyses (Table 4), including Degree (Appendix A), MCC (Appendix A), and MNC (Appendix A). Overlapping proteins according to the three topological analyses were ALYREF, HNRNPM, MRPS30, HNRNPL, PRPF19, MRPL10, MRPL23, EIF3K, RPS23, and RPS18 (Table 5).

#### 3.3.3. Outlier Proteins

The cut-off value obtained for LFC was 4.88. Two outliers were detected within the up-regulated DEP group, CHDH and ABCA5, and three in the down-regulated group: HEBP1, ATP2A3, and PLCD1 (see Appendix A for identification of the name of each protein with its abbreviation).

#### 3.3.4. Gene–Disease Association Analysis of DEPs

The Disease 2.0 database was used to perform gene-epilepsy association analysis of DEPs according to the literature. Genes with a confidence score > 2 in the three evidence channels were selected. The corresponding results are shown in Table 6.

In total, we detected 10 epilepsy-related genes that pass the filter between the Knowledge channel (Appendix A) and Text mining channel (Appendix A). No epilepsy-related genes were found in the Experimental channel (https://diseases.jensenlab.org/, accessed on 19 October 2022).

## 4. Discussion

The GASH/Sal model, a genetic model of audiogenic seizures from Salamanca, exhibits seizures triggered by sound with a short latency period. This suggests that there may not be enough time to synthesize new proteins in the epileptogenic nucleus, and thus pre-existing differences in protein expression compared to control hamsters may play a role in the susceptibility to seizures. Here, we aimed to study the protein profile in the IC of the GASH/Sal in comparison to control hamsters. Protein samples from the IC were processed for enzymatic digestion and then analyzed by MS in DIA mode. We identified 5254 proteins, of which 184 were DEPs, with 126 up-regulated and 58 down-regulated proteins. Moreover, 12 and 7 proteins were uniquely found in the GASH/Sal and in the control hamsters, respectively. Overexpressed DEPs are found almost exclusively in the nucleus, and are mainly involved in metabolic processes that affect the processing and modification of RNA and translation processes. On the other hand, underexpressed DEPs are involved in a greater number of molecular functions and biological processes than the overexpressed DEPs.

### 4.1. Methodological Discussion

This study employed next-generation label-free quantitative proteomics technology for the methodological analysis. In DIA mode, it can deliver unprecedented proteomic coverage while enabling accurate and highly repeatable quantification for large numbers of proteins per sample. The DIA analysis pipeline provides an ideal differential proteomic analysis or a proteomic quantification platform for large samples. This mode utilizes the latest high-resolution MS to simultaneously acquire peptide ion characteristics in mass and retention time-space. In comparison to the traditional technique of extracting a single ion for fragmentation analysis, the DIA mode uses an MS that is set to a wide precursor ion window to collect product ions sequentially. Thus, complete collection of all detectable protein peak information in the sample and highly reproducible analysis of large number of samples is achieved [45].

The bioinformatic analysis in this study relied on several commonly used databases, including the Gene Ontology (GO) and Eukaryotic Orthologous Groups (KOG) databases, which were consulted by default due to their established importance and usefulness in studying gene and protein function, pathways, and evolution.

### 4.2. Discussion of Results

#### 4.2.1. Identification of Group-Specific Proteins in GASH/Sal and Control Animals

We identified a total of seven unique proteins in the IC of control hamsters. Two of these proteins are involved in creatine metabolism: SLC6A8 (Solute Carrier Family 6 Member 8), a plasma membrane protein whose function is to transport creatine into and out of cells [46], and CKM (creatine kinase), involved in energy transduction. This last protein also regulates sodium–calcium exchanger activity [47], which plays an important role in maintaining intracellular Ca^2+^ homeostasis in cells. In addition, the protein RENBP (renin binding protein), previously reported by our group to be underexpressed in the IC of GASH/Sal compared to controls [32], was exclusively detected in the control animals in this study. Additionally, two other proteins unique to the control group were identified, SLC5A6 and FBXO11. SLC5A6, a sodium-dependent multivitamin transporter, has been associated with neurological disorders such as ataxia, developmental delay, and seizures under abnormal nutritional conditions [48]. FBXO11 (F-box only protein 11), on the other hand, is involved in the process of protein degradation and has been linked to lymphoproliferative disease when deficient [49], a pathology found in GASH/Sal under stress, due to the carriership of hamster polyomavirus (HaPyV) [27]. RASSF5 (Ras association domain-containing protein 5) is one of the proteins that, surprisingly, is overexpressed in mice hippocampus following kainic acid–induced seizure [50]. Our study identified another protein that is particularly relevant: the SHISA-9 protein encoded by the gene *LOC110339671*. This protein plays an important role in regulating the activity of glutamate receptors in the brain. It interacts with AMPA-type glutamate receptors and regulates their trafficking to the cell surface, which affects the strength of synaptic transmission. SHISA-9 is important for proper brain development and function, as well as for hearing, and its genomic region may influence synaptic plasticity, contributing to tinnitus perception [51]. Based on the findings of our study, the absence of SHISA-9 in the GASH/Sal model, in contrast to its presence in the control, highlights the potential of this protein as a promising candidate for further exploration of gene and protein expression, specifically concerning the altered hearing sensitivity and related morphological and molecular changes observed in the GASH/Sal model [34]. Supporting this, several studies have suggested neuropathological changes in the IC of the GASH/Sal model. Transcriptome analysis in this region of the GASH/Sal showed alterations in the transcriptional profile that might underlie changes in different biological processes involved in seizure occurrence and response [32,52]. The original lineage of the GASH/Sal model (the GPG/Vall hamster) showed decreased levels of GABA in the IC and anatomical changes such as a decrease in the volume and cell size of this cerebral area [53,54]. Our research group found that GABAergic system functioning is also impaired in the IC of the GASH/Sal strain [55], and hence it is likely that the GASH/Sal also exhibits those neuropathological changes. The proteomic bioinformatic analysis conducted in this study could aid in identifying crucial proteins for future investigations of gene and protein expression that could provide insight into the neuropathology of the IC.

Out of the 12 proteins that were exclusively detected in the IC of the GASH/Sal, there are four proteins that are particularly noteworthy. Diacylglycerol kinase (DGKZ) catalyzes the phosphorylation of diacylglycerol to produce phosphatidic acid [56] and remove 1-stearoyl-2-arachidonoylglycerol, the precursor of the endocannabinoid 2-arachidonoyl glycerol (2-AG) [57]. Interestingly, 2-AG itself is known to have anticonvulsive effects through activation of cannabinoid receptors [58], and mutant mice deficient in this enzyme had significantly fewer motor seizures and epileptic events compared with wild-type mice [59]. Therefore, the presence of this enzyme in the IC of GASH/Sal animals, but not in controls, could represent a risk factor in the triggering of seizures that occur in this hamster strain.

Two other proteins, ISG15 (ubiquitin-like protein ISG15) and PSMB9 (Proteasome subunit beta), are involved in the Type II interferon signaling pathway, widely expressed in cytokines associated with inflammation processes [60,61], and are likely involved in the inflammatory processes associated with convulsive crises.

Finally, it is important to emphasize the presence of the protein TRRAP (transformation/transcription domain-associated protein isoform X4) in the IC of the GASH/Sal, an adapter protein found in various multiprotein chromatin complexes with histone acetyltransferase activity, which gives a specific tag for epigenetic transcription activation [62]. Interestingly, this protein may play a role in the formation and maintenance of the auditory system [63], which might be relevant to the audiogenic seizures of the GASH/Sal model. *TRRAP* was added to the genetic epilepsy syndromes panel (https://panelapp.genomicsengland.co.uk/panels/402/, accessed on 20 October 2022).

To better understand whether certain proteins play a role in the susceptibility of the GASH/Sal experimental model for audiogenic seizures, further investigation is necessary to determine the significance of their higher levels in the epileptogenic nucleus compared to controls, especially regarding any possible modifications following seizures.

#### 4.2.2. Differentially Expressed Proteins in the IC of the GASH/Sal

From the 183 DEPs identified in our study, according to the information we reviewed (UniProtKB, Protein Data Bank, protein atlas, and NCBI), there is a large number of proteins, including their coding genes (more than 100), that we currently cannot correlate to alterations reported in the brain or to other changes previously described in the GASH/Sal.

Therefore, according to bibliographic and bioinformatics criteria, we selected a group of DEPs in non-bibliographic- and bibliographic-based approaches. Among the former, we included the results obtained by the PPI network analysis as well as those obtained by statistical methods and the outlier proteins. As part of the bibliographic-based approaches, we incorporated proteins from the Disease 2.0 database based on the criteria outlined in Section 2.7 of this manuscript.

##### PPI Network Analysis of DEPs

Among all the DEPs, we selected 10 overlapping hub proteins using the topological methods described in Section 2.7 as potential key proteins in the triggering of seizures in the GASH/Sal strain (Table 5).

ALYREF (THO Complex Subunit 4; DEP with 3.05 FC) is a nuclear protein that functions as a molecular chaperone. It is thought to regulate dimerization, DNA binding, and transcriptional activity of basic region-leucine zipper (bZIP) proteins (https://www.proteinatlas.org/ENSG00000183684-ALYREF, accessed on 20 October 2022). Among the various functions of ALYREF, as in other RNA-binding proteins, there is direct involvement in the export of miRNAs, because it plays critical roles in nuclear export [64,65]. Altered microRNA expression has been observed in the brain and blood of patients with various epilepsy disorders. MicroRNAs are the main regulators of gene expression- because single miRNAs impact multiple proteins with diverse effects within different pathways and networks. Changes both in levels and activity of miRNAs can induce profound effects on cellular function. Regulation of miRNA has high potential use in complex disorders like epilepsy, where numerous cellular pathways and processes may be affected simultaneously [66]. The understanding is that the miRNA–mRNA interactions will provide insights into epilepsy pathogenesis [67]. This is because another emerging therapeutic target is miRNA-small noncoding RNAs, which negatively regulate sets of proteins. Therefore, further studies on the ALYREF protein are regarded as having the potential to provide insight into epilepsy and potential novel therapeutic strategies.

PRPF19 (pre-mRNA processing factor 19; overexpressed DEP with 2.29 FC) is a ubiquitin-protein ligase that is a core component of several complexes, mainly involved pre-mRNA splicing and DNA repair, which is required for pre-mRNA splicing as a component of the spliceosome. As a core component of the spliceosome, PRPF19 participates in its assembly and its remodeling and is required for its activity (https://www.proteinatlas.org/ENSG00000110107-PRPF19, accessed on 21 October 2022). In addition, as part of the PSO4 core complex, this protein has been shown to participate in the DNA damage response. However, the specific role of the complex in DNA damage response pathways is still unclear [68]. Moreover, studies on mouse brain development suggest that PRPF19 serves as a molecular switch in governing neuron/glia differentiation, inhibiting neuronal differentiation while accelerating the differentiation of astrocytes. However, the neuronal function of human PRPF19 has rarely been investigated, and its involvement in human neurological diseases remains unexplored [69]. In conclusion, due to the vital role that PRPF19 plays in neural differentiation and in DNA damage and repair, it could be hypothetically related to alterations found in certain types of epilepsy. One example is epilepsy caused by focal cortical dysplasia, which originates from neurodevelopmental alterations.

HNRNPL (heterogeneous nuclear ribonucleoprotein L; overexpressed DEP with 2.02 FC) is a splicing factor binding to exonic or intronic sites and acting as either an activator or repressor of exon inclusion, exhibiting a binding preference for CA-rich elements. As part of a ribonucleoprotein complex that nucleates the complex on chromatin, it also negatively regulates the transcription of genes involved in neuronal differentiation (https://www.uniprot.org/uniprotkb/F1LQ48/entry, accessed on 21 October 2022) [70] or associated with alterations caused by trauma [71]. Both phenomena are closely interlinked with the presence of certain types of epilepsy, which could establish a potential relationship between this disorder and this protein. In addition to the former, the role of HNRNPL in both calcium metabolism [72] and potassium channel functioning [73] might be relevant for its relationship with seizures and epilepsy.

HNRNPM (heterogeneous nuclear ribonucleoprotein M; overexpressed DEP with 2.05 FC) is other abundant nuclear protein that binds to pre-mRNA and is another component of the spliceosome complex. RNA metabolism involves complex and regulated processes, some of which include transcription, intracellular transport, translation, and degradation. The involvement of RNA binding proteins in these processes remains mostly uncharacterized. Regarding brain function, loss of HNRNPM affects the physiological spine in vivo by impairing the morphology of the dendritic spines in the hippocampus [74]. Additionally, this protein directly binds to the 3’UTR of synaptophysin and PSD95 mRNAs, resulting in the stabilization of these mRNAs. This protein provides novel insight into the regulatory role of HNRNPM in neuronal structure and function [74]. This suggests that a potential alteration of the HNRNPM hippocampus levels might be associated with altered synaptic processes in epilepsy.

MRPL10 (39S mitochondrial ribosomal protein L10; overexpressed DEP with 2.06 FC) is part of the mammalian mitochondrial ribosomes and play roles in the mitochondrial respiratory chain. MRPL10 is encoded by nuclear genes, is synthesized in the cytoplasm, and then is transported to the mitochondria to be assembled into mitochondrial ribosomes. Thus, MRPL10 is necessary for mitochondrial protein synthesis and mitochondrial activity since the deletion of *MRPL10* reduces the mitochondrial activity and expression of the mitochondrial complex [75]. In addition, the deletion of this gene may negatively affect the adaptive metabolic response, cell proliferation, and cell survival due to inhibition of the kinase activity of cyclin-dependent kinase CDK1 [75]. Interestingly, GWAS analysis in humans showed association of the *MRPL10* gene with cytokines, which are also associated with epilepsy [76].

RPL23L (39S mitochondrial ribosomal protein L23; overexpressed DEP with 2.25 FC), also named MRPL23, is a structural constituent proteinic of ribosome and hence is involved in mitochondrial translation. This protein has been associated with the promotion of carcinoma metastasis and the occurrence of liver cancer [77]; however, to our knowledge no direct or indirect association with seizures or/and epilepsy has been reported.

MRPS30 (mitochondrial ribosomal protein S30; overexpressed DEP with 2.52 FC) helps in protein synthesis within the mitochondrion. Mutations in its coding gene have been associated with mitochondrial diseases, such us Leigh syndrome, a rare mitochondrial disorder in which patients can develop epilepsy [78]. Although epilepsy is a characteristic phenotypic feature of mitochondrial disorders, the prevalence of epilepsy is not equal among all patients with mitochondrial disorders [79]. In the case of patients with the type of syndrome or others, such as West syndrome and epilepsia partialis continua, the incidence of epilepsy is very low [80].

EIF3K (eukaryotic translation initiation factor 3 subunit K; overexpressed DEP with 2.11 FC) is the largest protein component of the eukaryotic translation initiation factor-3 complex, which is required for several steps in the initiation of protein synthesis [81]. This protein was found to be a potential biomarker of schizophrenia in a PPI network study of differentially expressed genes [82]; however, no association with epilepsy or seizures has been reported.

RPS23 (Ribosomal Protein S23; overexpressed DEP with 3.03 FC) is a large ribonucleoprotein complex and is a component of the ribosome responsible for the synthesis of proteins in the cell. This protein plays an important role in translational accuracy; missense mutations in its coding gene do not result in a reduction in the rate of mRNA translation but impair the accuracy of mRNA translation and render cells highly sensitive to oxidative stress [83]. The genes encode ribosomal proteins, such as *RPS23,* among others, which are enriched in biological processes of translation, translation elongation, and RNA processing in disc degeneration [84].

RPS18 (40S ribosomal protein S18; overexpressed DEP with 2.05 FC) is a component of the 40S ribosomal subunit, and alterations in its expression level could affect translation rate through an effect on ribosome subunit concentrations.

##### Protein Outliers

Furthermore, we included in our analysis the five statistical protein outliers; since although they do not present multiple interactions with other proteins, they have a significant representation (over- or underexpressed DEPs) and, therefore, they must have some differential significance in the functionality of the IC.

CHDH (choline dehydrogenase; overexpressed DEP with 6.68 FC) is a mitochondrial protein that regulates the concentrations of choline and glycine betaine in the blood and cells. Choline is important for regulation of gene expression as well as the biosynthesis of lipoproteins, membrane phospholipids, and the neurotransmitter acetylcholine [85]; thus, variations of CHDH can affect susceptibility to choline deficiency. Moreover, a population-based study showed that the metabolic oxidation of choline is related to the risk of developing breast cancer [86]. On the other hand, glycine betaine plays important roles as a primary intracellular osmoprotectant [87], and impairments in human CHDH activity have been associated with various pathologies, including male infertility, homocystinuria, metabolic syndrome, cardiovascular diseases, and breast cancer [87]. Despite all these data, at the moment, there is no association between this enzyme and epilepsy.

A type of ATP-binding cassette (ABC) transporter, part of a family of proteins associated with multidrug resistance, is another outlier overexpressed DEP. In particular, the ABCA5 (ATP-binding cassette sub-family A member 5; overexpressed DEP with 5.38 FC) was found to be overexpressed in the brain of patients with refractory epilepsy, suggesting an active drug efflux from brain [88]. Moreover, ABCA5 was implicated in the neuropathology associated with Alzheimer’s and Parkinson’s disease [89]. Alzheimer’s patients show specifically an increased expression of ABCA5 in hippocampal neurons, and amyloid-β-peptide levels are significantly reduced. Similar changes were observed with α-synuclein in neurons of the amygdala, where Parkinson’s disease patients showed higher levels of ABCA5. It is assumed that the overexpression of this protein is a protective response in these pathological conditions [90].

In the case of three outlier underexpressed DEPs, the protein levels of the first of these, PLCD1 (phosphoinositide phospholipase C; underexpressed DEP with −11.04 FC), and its downstream factor transient receptor potential channel 4 (TRPC4), which colocalize with glutamatergic and GABAergic neurons, were elevated in focal cortical dysplasia type II and tuberous sclerosis complexes, which are well-known causes of chronic refractory epilepsy in children [91]. Moreover, the lower levels of ATP2A3 (ATPase sarcoplasmic/endoplasmic reticulum Ca^2+^ transporting 3; underexpressed DEP with −9.95 FC), an ATPase that transports Ca^2+^ across membranes to the endoplasmatic reticulum to maintain a low cytoplasmic Ca^2+^ level in neurons of substantia nigra pars compacta in Parkinson’s disease, indicates a deficit in organelle function and Ca^2+^ sequestration [92]. Thus, increased levels of cytoplasmic Ca^2+^ due to lowered ATP2A3 levels could be detrimental to cells and cause degeneration [93]. In the case of HEBP1 (heme-binding protein 1; underexpressed DEP with −8.09 FC), Yagensky et al. [94] identified an increase of expression as a presymptomatic Alzheimer’s disease marker. This protein mediates heme-induced cytotoxicity via an apoptotic pathway, and interestingly, knockdown of *Hebp1* expression in neurons protects them from both heme and Ab42-induced apoptosis. Neurons lacking HEBP1 demonstrated resistance to apoptosis induced by hemin treatment. This resistance was due to the protein not being released into the cytosol, which prevented the activation of caspases 9 and 3/7, even though upstream activating events had occurred. These findings suggest that *HEBP1* may be involved in regulating the formation of apoptosomes, which are essential for cleaving procaspase 9 into its functional form.

##### Gene–Disease Association Analysis of DEPs

Finally, we included the potential functional significance of the 10 DEPs detected in the present work and directly related to epilepsy.

ATP6V1A (H^+^-transporting two-sector ATPase; underexpressed DEP with −2.17 FC) is part of a multimeric complex present in a variety of cellular membranes that acts as an ATP-dependent proton pump and plays a key role in pH homeostasis and intracellular signaling pathways. *ATP6V1A* variants, mainly clustering within the ATP synthase α/β family-nucleotide-binding domain, include early lethal encephalopathies and developmental encephalopathy in epilepsy [95,96,97]. Interestingly, several mutants of *ATP6V1A*, such as the p.Asp100Tyr, are characterized by reduced expression due to increased degradation; these mutations caused a defect in neurite elongation accompanied by loss of excitatory inputs, revealing that altered lysosomal homeostasis markedly affects neurite development and synaptic connectivity [95]. Furthermore, Persike et al. [98] used proteomics to determine the differential expression of proteins in the hippocampus of patients with mesial temporal lobe epilepsy (MTLE) compared to control samples, although in this case, ATP6V1A was up-regulated in comparison with our results. On the other hand, we also observed that ATP6V1B2, a subunit of the V-ATPase complex, was down-regulated. Pathogenic variants of *ATP6V1B2* have been associated with various epileptogenic phenotypes, including intellectual disability (ID), seizures, and/or DOORS syndrome (deafness, onychodystrophy, osteodystrophy, ID, and seizures) [99,100,101,102].

Alterations in the expression levels of BCAN (brevican core protein isoform X2; overexpressed DEP with 2.08 FC) and NCAN (neurocan core protein isoform X2; overexpressed DEP with 4.16 FC) have been reported in epileptic patients, as in the case of focal cortical dysplasia [103,104]. Epilepsies with a genetic basis can manifest early in life, and both *BCAN* and *NCAN* are believed to play crucial roles in terminal differentiation during development, as well as in the adult nervous system during postnatal development. Therefore, our data showing increased levels of expression of BCAN and NCAN in the GASH/Sal strain as compared with the control might explain an imbalance in the extracellular matrix composition that could be implicated in the origin or propagation of seizures.

The enzyme FAAH (fatty acid amide hydrolase), which was found to be overexpressed with a FC of 2.40 in the DEPs, is responsible for the breakdown of various primary and secondary fatty acid amides. This includes endocannabinoids such as anandamide, which have neuro-modulatory properties [105]. This enzyme, together with 2-arachydonoil glycerol, acts as a key regulator of glutamate and GABA release, providing different forms of synaptic plasticity in excitatory and inhibitory neurotransmission in several brain regions [106]. In summary, the overexpression of FAAH may lead to a reduced ability to protect against excessive neuronal activity, such as during epileptic seizures.

In addition, another DEP detected in our study is the GJA1 (gap junction protein; overexpressed DEP with 2.03 FC). This protein might be associated with epilepsy as it plays a significant role in phenomena related to cellular communications and provides a route for the diffusion of low-molecular-weight materials from cell to cell. In this regard, it is known that some of the proteins involved in cell-to-cell communication may represent potential targets in epilepsy treatment [107]. Consistently, our results showed an upregulation of GJA1, which might indicate an abnormal flow of molecules between cells and might potentially contribute to seizure genesis and propagation in the GASH/Sal.

In the case of the enzyme GLS (glutaminase; underexpressed DEP with −2.33 FC), which is involved in the conversion of glutamine into glutamate, Rumping et al. [108] identified two gene variants, a homozygous p.(Asp232Glufs*2) and compound heterozygous p.(Gln81* and Arg272Lys) in children with neonatal epileptic encephalopathy. These variants likely resulted in glutaminase deficiency, which led to an increase in glutamine levels in the affected children. Moreover, disturbed glutamine–glutamate shuttling is a known cause of epilepsy [109,110]. Thus, a loss of function in glutaminase can cause a neurometabolic disorder that leads to lethal early neonatal encephalopathy. Additionally, this disorder is likely to be associated with the epileptic phenotype observed in GASH/Sal hamsters. *GLS* gene is highly expressed in the brain and has a pivotal role in creating glutamate abundance in this organ, contrary to the systemic circulation, in which glutamine is the most abundant amino acid [111,112].

Another DEP associated with the glutamate metabolism is GLUL (glutamate-ammonia ligase; overexpressed DEP with 2.01 FC). This protein regulates the levels of toxic ammonia and converts neurotoxic glutamate into harmless glutamine, where loss of GLUL in astroglia is reported in the hippocampi of epileptic patients [113]. The two-fold increase in basal GLUL expression levels could indicate the need for greater availability of this enzyme by the GASH/Sal, a fact that can be correlated with elevated glutamate levels, which is widely described in epilepsy [114,115]. On the other hand, ammonium levels also modulate the occurrence of seizures in myoclonic epilepsy and various other related conditions [116,117]. GLUL is directly related to the detoxification process of both compounds mentioned above, and both could be found increased during and after convulsive crises, such as those described in the GASH/Sal model.

Furthermore, GSR (glutathione reductase; overexpressed DEP with 2.46 FC) is a central enzyme of cellular antioxidant protection that is closely linked to glutathione metabolism, reducing oxidized glutathione disulfide (GSSG) to the sulfhydryl form GSH, which is a valuable cellular antioxidant. In this sense, GSR maintains high levels of reduced glutathione in the cytosol. As occurred in the GLUL enzyme and other compounds involved in the metabolism of glutathione, an increase of GSR levels may also indicate the intensification of oxidative stress machineries as extensively reported in seizures [118,119]. Therefore, we hypothesized that elevated levels of the GSR enzyme could indicate the occurrence of compensatory mechanisms that help reduce harmful elements in the cell. This could be relevant to the seizure susceptibility of GASH/Sal, since noxious elements are generated within neurons, mainly due to the excitatory period they experience.

A type of heat shock protein (HSP) was detected within the DEPs in our study and related with epilepsy: HSPA4 (heat shock 70 kDa protein 4; underexpressed DEP with −2.20 FC). The HSP provides a first line of defense against accumulation of misfolded proteins as well as accomplishing other neuroprotective effects, e.g., inhibition of apoptosis, protection of cytoskeletal structures and immune modulation in stress conditions [120]. Interestingly, it has also been reported that HSPA4, during the latency phase following status epilepticus, shows a significant upregulation in subregions of the temporal lobe crucial for the development of TLE [121] and in a pilocarpine model of this type of epilepsy [122]. It is, therefore, plausible that cytomolecular changes in the neuronal circuitry during epileptogenesis lead to increased endoplasmic reticulum stress with following upregulated expression of HSPH4, which would exert protective effects as a response to stress factors [123]. Low levels of this enzyme would make GASH/Sal more vulnerable to convulsive episodes. However, it would be necessary to study the modifications of this protein after the convulsive crises.

Finally, in the present work we detected several DEPs linked to the functioning of glial cells, such as the myelin basic protein (MBP; overexpressed DEP with 2.09 FC), the most abundant protein component of the myelin membrane in the CNS. MBP has a key role in both myelin formation and its stabilization; however, studies that discuss the link between MBP and epilepsy are scarce. Thus, for example, children with seizures had normal MBP levels in cerebral spinal fluid, similar to the controls; that is, for individual patients, MBP is of little value as a prognostic indicator [124]. Another study reported that up-regulation of MBP protein levels in zinc diet-treated animals with seizures may represent a compensatory mechanism for neuronal membrane damage and repair [125]. Nevertheless, a decrease of MBP levels in the epileptic foci of patients suffering from intractable epilepsy is reported [126]. We are currently unable to provide a plausible explanation for the result observed in MBP. Thus, it is essential to conduct further studies on the glial population in GASH/Sal hamsters to obtain additional information regarding this under-explored field in this experimental model of epilepsy.

## 5. Conclusions

The combination of ESI-orbitrap MS with comprehensive bioinformatic analysis has proven to be an effective tool for characterizing proteomic shifts in the IC of GASH/Sal hamsters compared to controls. In this study, we successfully identified 126 overexpressed and 58 underexpressed differential proteins, 10 of which are directly related to epilepsy. We also discovered 12 proteins unique to GASH/Sal hamsters and seven unique to the control group. Our bioinformatic analysis allowed us to identify 25 potential hub proteins, many of which are not directly related to epilepsy. These findings significantly increase our understanding of this pathology and provide a framework for further studies on the mechanisms underlying the proteomic changes in the IC of GASH/Sal hamsters, potentially leading to the use of this audiogenic seizure model in the development of novel therapeutic approaches for epilepsy.

## Figures and Tables

**Figure 1 diagnostics-13-01048-f001:**
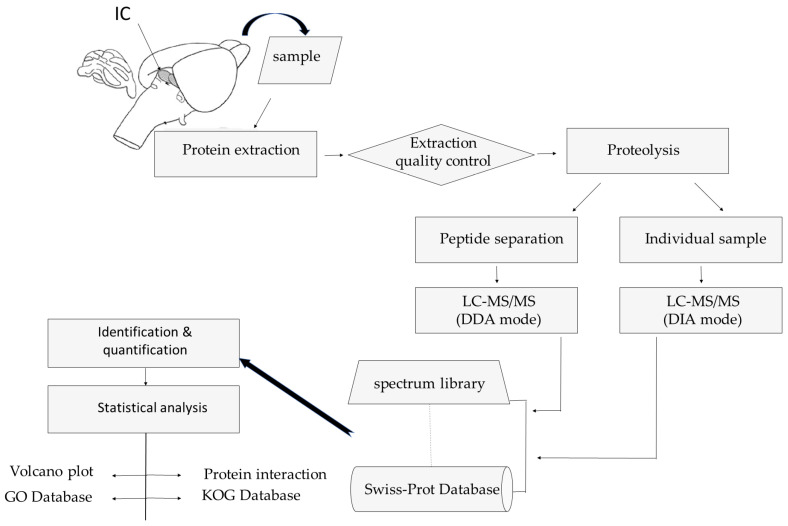
Experimental workflow. The main experimental steps are shown in the above figure. Abbreviations: IC—inferior colliculus (sample); LC-MS/MS—liquid chromatograph followed by tandem mass spectrometry; DDA—data-dependent acquisition; DIA—data-independent acquisition; GO—gene ontology; KOG—eukaryotic orthologous groups of proteins.

**Figure 2 diagnostics-13-01048-f002:**
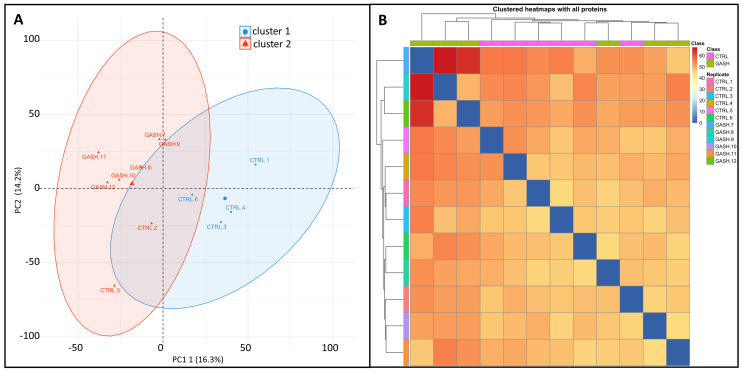
Principal Component Analysis (PCA) and clustering with all proteins. (**A**) Scatter plot of PCA of protein quantitative values from each sample in each group. k-means clustering PCA. The *X*-axis is the first principal component, and the *Y*-axis is the second principal component. (**B**) HeatMap of sample correlation analysis. Both *X* and *Y* axes represent samples. The color represents the correlation coefficient (the red color represents the higher correlation; the blue color represents the lower correlation).

**Figure 3 diagnostics-13-01048-f003:**
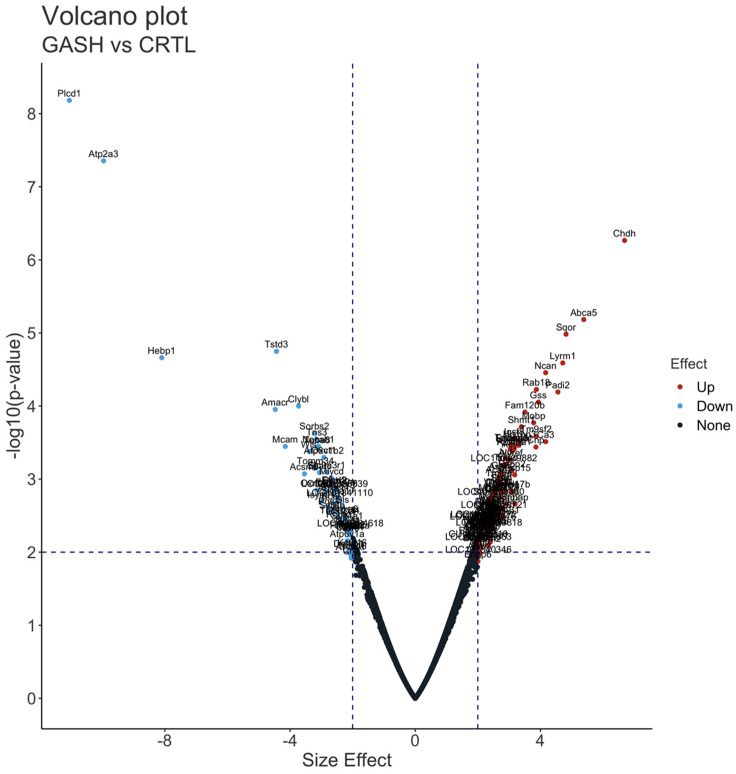
Volcano plot of DEPs. The *X*-axis of the graph is the Cohen’s effect size, and the *Y*-axis is the corresponding −log10 (*p*-value). The red dot indicates significantly up-regulated proteins, the blue dot indicates significantly down-regulated proteins in the IC of the GASH/Sal vs. control hamster, and the black dots indicate proteins without significant change. The gene names that encode the significant proteins are shown on the graph, and for better visualization, an interactive version of Figure 3 is provided in the Appendix A.

**Figure 4 diagnostics-13-01048-f004:**
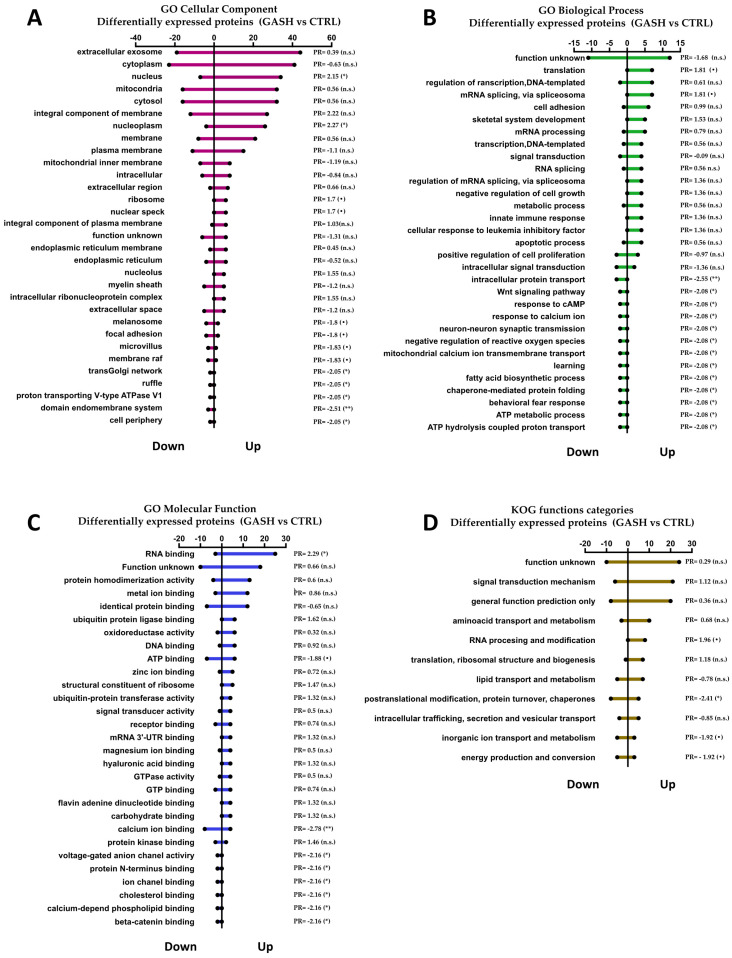
DEPs: GO analysis and KOG functions. The *X*-axis represents the number of DEPs up- and down-regulated in the IC in the GASH/Sal (GASH) vs. control (CRTL) hamster number of proteins, and the *Y*-axis represents the different annotation entries. The *Chi*-squared test assessed the relationship between the groups (CTRL/GASH) and the annotation entries. (**A**) GO Cellular component; *Chi*-squared *p*-value = 0.02649. (**B**) GO Biological process; *Chi*-squared *p*-value = 0.0065. (**C**) GO Molecular functions; *Chi*-squared *p*-value = 0.02649. (**D**) KOG functions; Chi-squared *p*-value = 0.16542. The PR (Pearson residual) shows the significance of variables between groups: • (90–95% confidence); * (95–97.5% confidence); ** (97.5–99% confidence); n.s. (no significant differences).

**Figure 5 diagnostics-13-01048-f005:**
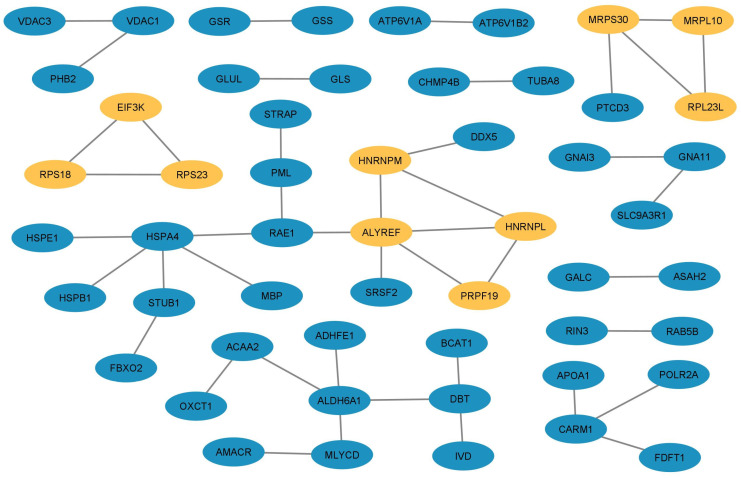
Protein–Protein Interaction (PPI) network analysis. The PPI network (with a String-interactions short-score ≥ 2) included 53 nodes and 44 edges. Orange nodes represent overlapping proteins between three topological analyses.

**Table 1 diagnostics-13-01048-t001:** Characterization of missing values (NA) in each experiment.

Replicate	Group	Non-NA Proteins	NA Proteins
**1**	CONTROL	5175	79
**2**	CONTROL	5174	80
**3**	CONTROL	5195	59
**4**	CONTROL	5146	108
**5**	CONTROL	5161	93
**6**	CONTROL	5162	92
**1**	GASH/Sal	5192	62
**2**	GASH/Sal	5156	98
**3**	GASH/Sal	5184	70
**4**	GASH/Sal	5179	75
**5**	GASH/Sal	5193	61
**6**	GASH/Sal	5149	105

**Table 2 diagnostics-13-01048-t002:** Proteins exclusive to the IC of control hamsters.

Accession	Gene	Name
A0A1U7QBG8	*Renbp*	Renin Binding Protein
A0A1U7QIW7	*Rassf5*	Ras Association Domain containing protein 5
A0A1U7R5G9	*Ckm*	Creatine kinase
A0A1U8BZ60	*Slc5a6*	sodium-dependent multivitamin transporter
A0A1U8CDT5	*Fbxo11*	F-box protein 11
A0A1U8CS96	*Slc6a8*	Transporter (Solute Carrier Family 6 Member 8)
A0A3Q0CFS1	*LOC110339671*	protein shisa-9

**Table 3 diagnostics-13-01048-t003:** Proteins exclusive to the IC of the GASH/Sal strain.

Accession	Gene	Name
A0A1U7Q366	*Dgkz*	Diacylglycerol kinase
A0A1U7Q7Q4	*Apc*	adenomatous polyposis coli protein isoform X5
A0A1U7Q9R8	*Faim2*	protein lifeguard 2 isoform X1
A0A1U7QE28	*Fam151b*	protein FAM151B isoform X1
A0A1U7QKE9	*Trim21*	E3 ubiquitin-protein ligase TRIM21
A0A1U7QSF3	*Dbr1*	lariat debranching enzyme
A0A1U7QVI9	*Psmb9*	Proteasome subunit beta
A0A1U7QZ55	*Nosip*	Nitric oxide synthase-interacting protein
A0A1U8C8Y2	*Trrap*	transformation/transcription domain-associated protein isoform X4
A0A1U8CAF8	*Isg15*	ubiquitin-like protein ISG15
A0A3Q0CKC8	*LOC106021169*	LOW QUALITY PROTEIN: cytochrome c oxidase subunit 6C-2
Q60543	*Serpina6 (CBG)*	Corticosteroid-binding globulin

**Table 4 diagnostics-13-01048-t004:** Hub proteins ranked with a score ≥ 2 in the *cytoHubba* plugin of Cytoscape with the three topological analyses: Degree, maximal clique centrality (MCC) and maximum neighborhood component (MNC). The 10 proteins common to the three algorithms are presented in bold.

Rank Methods in CytoHubba		
Degree	MCC	MNC
HSPA4	**ALYREF**	**ALYREF**
ALYREF	HSPA4	**HNRNPL**
ALDH6A1	ALDH6A1	**PRPF19**
**HNRNPM**	**HNRNPL**	**HNRNPM**
RAE1	**HNRNPM**	**MRPL10**
CARM1	RAE1	**RPL23L**
**MRPS30**	CARM1	**MRPS30**
DBT	**MRPS30**	**EIF3K**
**HNRNPL**	DBT	**RPS23**
**PRPF19**	**PRPF19**	**RPS18**
**MRPL10**	**MRPL10**	
**RPL23L**	**RPL23L**	
VDAC1	VDAC1	
ACAA2	ACAA2	
PML	PML	
**EIF3K**	**EIF3K**	
**RPS23**	**RPS23**	
**RPS18**	**RPS18**	
STUB1	STUB1	
GNA11	GNA11	
MLYCD	MLYCD	

**Table 5 diagnostics-13-01048-t005:** Summary of functions of 10 overlapping proteins of Table 4.

Summary of Functions of Hub Proteins	
Protein	Name of Proteins and Functions
ALYREF	THO Complex Subunit 4. Regulate dimerization, DNA binding, and transcriptional activity
EIF3K	Eukaryotic translation initiation factor 3 subunit K
HNRNPL	heterogeneous nuclear ribonucleoprotein L isoform. Play a major role in the formation, packaging, processing, and function of mRNA.
HNRNPM	heterogeneous nuclear ribonucleoproteinM isoform. Appear to influence pre-mRNA processing and other aspects of mRNA metabolism and transport.
MRPL10	39S Mitochondrial ribosomal protein L10, X1.Help in protein synthesis within the mitochondria
MRPS30	39S Mitochondrial ribosomal protein S30. Help in protein synthesis within the mitochondria
PRPF19	Pre-mRNA-processing factor 19. Essential for cell survival and DNA repair
MRPL23	Mitochondrial Ribosomal Protein L23. Help in protein synthesis within the mitochondria
RPS18	40S ribosomal protein S18. Protein that is a component of the 40S subunit.
RPS23	40S ribosomal protein S23. Protein that is a component of the 40S subunit.

**Table 6 diagnostics-13-01048-t006:** Epilepsy related genes with a confidence score > 2 in the Disease 2.0 database.

Gene-Disease Association Analysis of DEPs				
	Gene Name	Disease Identifier	Disease Name	Confidence Score
knowledge chanel	*ATP6V1A*	DOID:1826	Epilepsy	4
	*GLS*	DOID:1826	Epilepsy	4
Text mining channel	*GLUL*	DOID:1826	Epilepsy	2.507
	*GJA1*	DOID:1826	Epilepsy	2.483
	*MBP*	DOID:1826	Epilepsy	2.404
	*GLUL*	DOID:3328	Temporal lobe epilepsy	2.37
	*GLUL*	DOID:2234	Focal epilepsy	2.34
	*GSR*	DOID:1826	Epilepsy	2.23
	*FAAH*	DOID:1826	Epilepsy	2.212
	*HSPA4*	DOID:1826	Epilepsy	2.142
	*NCAN*	DOID:1826	Epilepsy	2.067
	*GJA1*	DOID:2234	Focal epilepsy	2.018
	*GJA1*	DOID:3328	Temporal lobe epilepsy	2.006
	*BCAN*	DOID:1826	Epilepsy	2.003

## Data Availability

The mass spectrometry proteomics data have been deposited to the ProteomeXchange Consortium via the PRIDE [127] partner repository with the dataset identifier PXD039755.

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
