# Peer review of "Proteomic and Bioinformatic Tools to Identify Potential Hub Proteins in the Audiogenic Seizure-Prone Hamster GASH/Sal"

_diagnostics, 2023, doi:10.3390/diagnostics13061048_

Round 1

Reviewer 1 Report

Interesting and comprehensive. Some key methods issues need clarification as below:

The age range of 2-4 months is quite broad and ranges from adoles to full adult. pls comment.

Choice of all male hamsters. Pls comment.

The description of how they made the protein lysis solution seemed reasonable, but the last sentence of 2.2 was too brief.  Which trypsin was used, what enzyme: substrate ratio, what digestion temp and time?  What is a Strata X column and how was it used?  Were disulfide bonds reduced and the cysteines alkylated with something?

Section 2.3: its not clear how these 10 high pH fractions relate to the samples that are later subjected to DIA analysis.  Presumably, the latter are unfractionated peptides from individual hamsters.     Section 2.4 should state what sort of trap column was used, and how samples were loaded to the trap.  For the last paragraph in 2.4 (For DIA analysis...) they should state what the cycle time is for their method (ie, how many data points are acquired across a chromatographic peak), and what was the precursor isolation width for each DIA spectrum.   Figure 1 diagram contains the word 'Proteolisis' but it should be 'Proteolysis'.  Figure 1 seems to suggest that after Proteolisis (sic) Peptide separation occurs and precedes LC-MS/MS (DIAmode).  This suggests that all samples were individually fractionated to make 10 fractions per hamster, so 120 LCMSMS runs were done (12 hamsters x 10 fractions).  This doesn't seem right, so I'm not sure that Figure 1 helps me to understand what they did.   One thing that seems to be missing here is a description of how the proteins were inferred from the peptide identifications.  How many peptides were needed to say a particular protein was present?  What do they do with peptides found in more than one protein?  How do they group proteins that are indistinguishable from each other (eg, three proteins with the same two peptides should be grouped as one)

Author Response

Ref: Manuscript ID: diagnostics-2219640
Title: Proteomic and bioinformatic tools to identify potential hubs proteins in the audiogenic seizure-prone hamster GASH/Sal
Journal: Diagnostics: section: Pathology and Molecular Diagnostics. Special Issue of Diagnostics: State-of-the-Art Research on Epilepsy

Dear Ms Iulia Radulescu,

We would like to thank you again for the effort and time that the editors and reviewers have made to highlight the results of this manuscript. We have taken the Reviewer’s recommendations seriously and modified the manuscript accordingly to his/her suggestions. After receiving suggestions from the reviewers, the manuscript was revised with the specific aim of improving its clarity and ease of reading.

We agree with many of the changes proposed by Reviewers, and we think they are worthwhile. We will reply in detail to each of their comments below.

As many changes have been made, we have sent the manuscript with the changes highlighted, as requested, and a new manuscript with the changes in red for easier reading. Please, send it to the referees.

Yours sincerely,

Dolores E. López

On behalf of all the authors

Addressing comments and concerns from Reviewer 1.

- Reviewer 1
  - Interesting and comprehensive. Some key methods issues need clarification as below:

QUESTIONS

  1. The age range of 2-4 months is quite broad and ranges from adoles to full adult. Pls comment.

Answer:

Thanks a lot for all your useful comments. As we wrote in the M&M section, we selected animals of 2-4 months because in this period the GASH/Sal exhibits the maximum susceptibility to seizures. All the previous experiments carried out in our laboratory used animals in that age range. It cannot be further refined as the commercial houses that supply us with the controls send us animals with ages in that range and we choose those with similar ages among the animals from the colony. Using a consistent age range for experimental animals allows for comparison across various studies and experiments.

  1. Choice of all male hamster. Pls comment

Answer:

In recent years, all the studies carried out in the laboratory, anatomical, physiological...etc), have been carried out with male animals to avoid variability due to hormonal factors. Using both male and female animals in our research would require additional resources and more animals, which is currently not feasible. However, we plan to expand our research in the future to include female animals as well, in order to investigate potential sex differences.

  1. The description of how they made the protein lysis solution seemed reasonable, but the last sentence of 2.2 was too brief. Which trypsin was used, what enzyme: substrate ratio, what digestion temp and time?

Answer:

2.5μg of Trypsin enzyme (Trypsin, Mass Spec Grade) with the ratio of protein: trypsin = 40:1 at 37°C for 4 hours.

  1. What is a Strata X column and how was it used? Were disulfide bonds reduced and the cysteines alkylated with something?

Answer:

Strata X SPE desalting column (Strata X 33μm polymeric reversed-phase column; Phenomenex, Torrance, CA, USA). Disulfide bonds were reduced with DTT (DL-Dithiothreitol) with the final concentration of 10mM; water bath at 56°C for 1 hour. Cysteines were alkylated with IAM (iodoacetamide) at a final concentration of 55mM. and place in a dark room for 45 minutes

  1. Section 2.3: its not clear how these 10 high pH fractions relate to the samples that are later subjected to DIA analysis. Presumably, the latter are unfractionated peptides from individual hamsters.

Answer:

The referee is right. The ten high pH fractions were subjected to Data-dependent Acquisition (DDA) and the unfractionated to DIA. On one side, each of the ten high pH fractions (fractioned peptide sample) was subjected to DDA analysis to construct the spectral library. A spectral library collects all detectable non-redundant, high-quality peptide information (MS/MS spectra) of the sample that can be used as a peptide identification template for subsequent data analysis. It contains fragment ion intensity and retention time that characterize the peptide spectrum. Conversely, an unfractionated peptide sample was subjected to Data independent acquisition (DIA, also called SWATH) mode that utilizes the latest high-resolution mass spectrometer to acquire peptide ion characteristics in mass and retention time-space simultaneously. Compared to the traditional technique of extracting a single ion for fragmentation analysis, the mass spectrometer is set to a wide precursor ion window in DIA mode to collect product ions. Thus, the complete collection of all detectable protein peak information in the sample and high-reproducible analysis of many samples is achieved.

Identification and quantification of peptides and proteins were obtained from DDA spectral library by deconvolution of the DIA data.

  1. It should state what sort of trap column was used and how samples were loaded into the trap

Answer:

Separation was carried out by a Thermo UltiMate 3000 UHPLC liquid chromatograph. The sample was first enriched in the trap column and desalted, and then entered a tandem self-packed C18 column (150μm internal diameter, 1.8μm column size, 35cm column length) and separated at a flow rate of 500nL/min by the following effective gradient: 0~5 minutes, 5% mobile phase B (98% ACN, 0.1% FA); 5~120 minutes, mobile phase B linearly increased from 5% to 25%; 120~160 minutes, mobile phase B rose from 25% to 35%; 160~170 minutes, mobile phase B rose from 35% to 80%; 170~175 minutes, 80% mobile phase B; 175~180 minutes, 5% mobile phase B.

  1. For the last paragraph in 2.4 (For DIA analysis...) they should state what the cycle time is for their method (ie, how many data points are acquired across a chromatographic peak), and what was the precursor isolation width for each DIA spectrum.

Answer:

For DIA analysis, LC separated peptides were ionized by nanoESI and injected to tandem mass spectrometer Q-Exactive HF X (Thermo Fisher Scientific, San Jose, CA) with DIA (data-independent acquisiton) detection mode as the following settings: ion source voltage 1.9 kV; MS scan range 400~1,250 m/z; MS resolution 120,000, MIT 50 ms; 400~1,250 m/z was equally divided to 45 continuous windows MS/MS scan. MS/MS collision type HCD, MIT (maximal injection time) was auto mode. Fragment ions were scanned in Orbitrap, MS/MS resolution 30,000, collision energy was distributed mode: 22.5, 25, 27.5, AGC was 1E6.

  1. Figure 1 diagram contains the word 'Proteolisis' but it should be 'Proteolysis'.

Answer:

Changed as requested.

  1. Figure 1 seems to suggest that after Proteolisis (sic) Peptide separation occurs and precedes LC-MS/MS (DIAmode). This suggests that all samples were individually fractionated to make 10 fractions per hamster, so 120 LCMSMS runs were done (12 hamsters x 10 fractions). This doesn't seem right, so I'm not sure that Figure 1 helps me to understand what they did.

Answer:

Thank you for your positive feedback on Figure 1. Please, note that Figure 1 has been rearranged. To learn more about the details of the figure, please refer to Section 2.3 for an explanation.

  1. One thing that seems to be missing here is a description of how the proteins were inferred from the peptide identifications.

Answer:

The Andromeda search engine within MaxQuant identified DDA data, and identification results were used for spectral library construction. The mProphet algorithm was used for large-scale DIA data to complete analytical quality control, thus obtaining many reliable quantitative results.

  1. How many peptides were needed to say a particular protein was present?

Answer:

Every quantification experiment obviously has a first phase focused on the identification of the peptides and proteins generated in the digestion with trypsin. The number of unique (ie, different) peptides identified for each protein is highly variable, from 1 to several tens. This is usually associated with the relative abundance of the protein in the sample, that is, the most abundant proteins tend to generate more identified peptides.

One percent false discovery rate (1% FDR) using Target-Decoy database search was used for peptide validation.

Only protein groups with at least one unique peptide identified were considered.

The protein identification analysis was made by the BGI company (Shenzhen, China). (https://www.bgi.com/global/service/protein-identification-2)

  1. What do they do with peptides found in more than one protein?

Answer:

They are only used as a reference if it is not the unique peptide.

  1. How do they group proteins that are indistinguishable from each other (eg, three proteins with the same two peptides should be grouped as one).

Answer:  

If the peptides of two proteins are exactly the same, those proteins are classified as part of the same set.

Global response to Reviewer 1’s comments.

With all the suggestions made by the referee, we have modified the M&M section. Specially, we included major modifications regarding revision in sections 2.2 (questions 3 and 4), 2.4 (questions 5, 8 and 9), 2.5 (questions 10, 11).

As noticed in the revised manuscript, we added one new section (2.5 Data Analysis for protein identification) in M&M section. We are grateful to Reviewer 1 for his/her contribution to improving the manuscript by helping us to provide more accurate information regarding the materials and methods used.

Reviewer 2 Report

The authors describe a proteomic approach to identify differentially expressed gene in the inferior colliculus of the epilepsy model Genetic Audiogenic Seizure Hamster. I have the following critical remarks.

1. What about neuropathological changes in the IC of the epileptic hamsters? Is there any evidence that the observed changes might be cell composition related? That should be discussed in adequate detail.

2. Have been changes in cell counts of neurons and glial cells detected in IC of epileptic hamsters?

3. It is not clear which proteins are shown in Figs. 3-5, and why this type of analysis was performed for all proteins.

4. The tables with exclusive proteins are mixed up in the text (lines 321-322). Moreover, it remains obscure, why Ckm should be expressed in the IC of control hamsters only.

5. A control experiment from a brain region being not involved in seizure activity would be helpful to exclude potential sampling artifacts.

Author Response

Ref: Manuscript ID: diagnostics-2219640
Title: Proteomic and bioinformatic tools to identify potential hubs proteins in the audiogenic seizure-prone hamster GASH/Sal
Journal: Diagnostics: section: Pathology and Molecular Diagnostics. Special Issue of Diagnostics: State-of-the-Art Research on Epilepsy

Dear Ms Iulia Radulescu,

We would like to thank you again for the effort and time that the editors and reviewers have made to highlight the results of this manuscript. We have taken the Reviewer’s recommendations seriously and modified the manuscript accordingly to his/her suggestions. After receiving suggestions from the reviewers, the manuscript was revised with the specific aim of improving its clarity and ease of reading.

We agree with many of the changes proposed by Reviewers, and we think they are worthwhile. We will reply in detail to each of their comments below.

As many changes have been made, we have sent the manuscript with the changes highlighted, as requested, and a new manuscript with the changes in red for easier reading. Please, send it to the referees.

Yours sincerely,

Dolores E. López

On behalf of all the authors

Addressing comments and concerns from Reviewer 2.

-Reviewer 2

The authors describe a proteomic approach to identify differentially expressed gene in the inferior colliculus of the epilepsy model Genetic Audiogenic Seizure Hamster. I have the following critical remarks.

Answer: Thanks a lot for your useful review and comments. Following your suggestion, the manuscript has been revised in order to improve the wording.

QUESTIONS

  1. What about neuropathological changes in the IC of the epileptic hamsters? Is there any evidence that the observed changes might be cell composition related? That should be discussed in adequate detail.

Answer:

There have been studies examining the neuropathological changes that occur in the inferior colliculus and the auditory pathway of epileptic hamsters. The inferior colliculus is a structure in the midbrain that is involved in processing auditory information, and it is critical for generation and propagation of audiogenic seizure. One study from our research group found that there are morphological and functional alterations of bottom-up auditory inputs to the inferior colliculus (Sánchez-Benito et al., 2020). This study demonstrated aberrant glutamatergic transmission in the flow of sound processing from the inner ear to the cochlear nucleus to the inferior colliculus. In addition, the auditory brain recordings corresponding to the inferior colliculus (wave IV) showed a reduction in the amplitude (see Fig. 1 in Sánchez-Benito et al., 2020). In addition, the transcriptome analysis in the inferior colliculus of the GASH/Sal model showed alterations in the transcriptional profile that might underlie changes in different biological processes involved in seizure occurrence and response, and indirectly contributing to the susceptibility to audiogenic seizures (Díaz-Rodríguez et al., 2020; doi: 10.3389/fnins.2020.00508; Damasceno et al., 2020: doi: 10.3389/fneur.2020.00033). Furthermore, the original lineage of the GASH/Sal model (the GPG/Vall hamster) showed decreased levels of GABA in the inferior colliculus (Fuentes-Santamaría et al., 2008: doi: 10.1016/j.eplepsyres.2008.02.003) and anatomical changes such as a decrease in the volume and cell size the inferior colliculus (Fuentes-Santamaría et al., 2005; https://doi.org/10.1111/j.1528-1167.2005.68104.x). Our research group found that the GABAergic system functioning is impaired in the GASH/Sal strain (Prieto-Martín et al., 2017 http://dx.doi.org/10.1016/j.yebeh.2015.05.025), and hence, it is likely that the GASH/Sal also exhibits those neuropathological changes in the IC. Overall, these studies suggest structural and functional changes in the inferior colliculus of the GASH/Sal as it might be common in other rodent models of audiogenic seizures (Damasceno et al., 2020).

We totally agree with Reviewer 2 that this important information should be discussed and correlated with our results in the current manuscript. Interestingly, our results showed the absence of SHISA-9 in the GASH/Sal model but its presence in the control. The protein SHISA-9 is a transmembrane protein that is primarily expressed in the brain and is involved in regulating the activity of glutamate receptors. This protein may affect synaptic plasticity and lead to auditory alterations (Bhatt et al., 2022. https://doi.org/10.1038/s41598-022-26413-6) like those observed in the GAHS/Sal model.

Following the reviewer’s suggestion, we have modified the discussion section of the manuscript to add this important information. Please, notice that we have added new references in the manuscript.  

We have included the following information of SHISA-9 protein in the revised manuscript as described below:

Previous version of the manuscript: Finally, the function of LOC110339671 (protein shisa-9) is unknown.

Revised version of the manuscript: Our study identified another protein that is particularly relevant to discuss: the SHISA-9 protein encoded by the gene LOC110339671. This protein plays an important role in regulating the activity of glutamate receptors in the brain. It interacts with AMPA-type glutamate receptors and regulates their trafficking to the cell surface, which affects the strength of synaptic transmission. SHISA-9 is important for proper brain development and function, as well as for hearing, and its genomic region may influence synaptic plasticity contributing to tinnitus perception [52]. Based on the findings of our study, the absence of SHISA-9 in the GASH/Sal model, in contrast to its presence in the control, highlights the potential of this protein as a promising candidate for further exploration of gene and protein expression, specifically concerning the altered hearing sensitivity and related morphological and molecular changes observed in the GASH/Sal model [34]. Supporting this, several studies have suggested neuropathological changes in the IC of the GASH/Sal model. Transcriptome analysis in this region of the GASH/Sal showed alterations in the transcriptional profile that might underlie changes in different biological processes involved in seizure occurrence and response [32, 53]. The original lineage of the GASH/Sal model (the GPG/Vall hamster) showed decreased levels of GABA in the IC and anatomical changes such as a decrease in the volume and cell size of this cerebral area [54, 55]. Our research group found that the GABAergic system functioning is also impaired in the IC of the GASH/Sal strain [56], and hence it is likely that the GASH/Sal also exhibits those neuropathological changes. The proteomic bioinformatic analysis conducted in this study could aid in identifying crucial proteins for future investigations of gene and protein expression that could provide insight into the neuropathology of the IC.

  1. Have been changes in cell counts of neurons and glial cells detected in IC of epileptic hamsters?

Answer:

As previously mentioned, it is likely that the changes observed in the gene and protein expression of the inferior colliculus in the GASH/Sal model correspond to alterations in the number of neurons and glial cells. Several studies, which we have included in the discussion section of the manuscript, have reported morphological changes in the inferior colliculus and other regions of the auditory pathway in this model [34]. While we have not yet performed stereological analyses to investigate this aspect further, we have obtained preliminary 3D reconstruction data using Neurolucida software that indicate a reduction in the volume of the GASH/Sal inferior colliculus compared to controls. We plan to publish these results in the future once they are finalized. We appreciate the constructive feedback from Reviewer 2 and hope that the modifications we have made to the discussion section of the manuscript adequately address their concerns.

  1. It is not clear which proteins are shown in Figs. 3-5, and why this type of analysis was performed for all proteins.

Answer:

We would like to express our gratitude to Reviewer 2 for bringing an important matter to our attention. Initially, we had performed functional annotation of all identified proteins in our DIA analysis, using information from GO, KEGG, and KOG databases, as a standard approach. However, considering our subsequent analysis which involved an enrichment analysis of differentially expressed proteins, we agree with Reviewer 2 that this is not necessary and could potentially confuse the reader. Therefore, we have decided to remove Figs. 3-5, along with text and section 3.2, from the original manuscript for the sake of clarity.

  1. The tables with exclusive proteins are mixed up in the text (lines 321-322).

Answer:

We agree with Reviewer 2. The manuscript has been changed accordingly.

  1. Moreover, it remains obscure, why Ckm should be expressed in the IC of control hamsters only.

Answer:

Proteins detected only in one of the two experimental groups were found to be within the dynamic range of the detector used in the mass spectrometry analysis. Any signals below this range were not reported, which may explain why the Ckm protein was not detected in the GASH/Sal group. Therefore, our results should be interpreted with this in mind, and should only be viewed as a screening approach to identify proteins with potential involvement in seizure generation and propagation. It is important to note that differentially expressed proteins identified through our approach may not necessarily be involved in these processes. To verify this, we plan to conduct further studies and design new experiments using different techniques.

  1. A control experiment from a brain region being not involved in seizure activity would be helpful to exclude potential sampling artifacts.

Answer:

We appreciate the insightful comment and suggestion from Reviewer 2. As stated earlier, our study aimed to identify proteins that could potentially contribute to the development and spread of seizures in GASH/Sal using bioinformatics for global screening. The ultimate goal was to identify potential protein candidates for further validation through other techniques. While Reviewer 2's proposal to repeat the analysis in a tissue unrelated to seizure generation is intriguing, it would require a different study design. Firstly, we would need to choose an appropriate brain region, which should not only be non-epileptic but also excluded from repeated seizure stimulation (kindling). Thus, there are many brain regions that could be included in the study, posing additional interesting questions. Afterward, we would need to analyze the correlation between this region and the results obtained from the epileptogenic area (inferior colliculus). We believe that such an analysis is beyond the scope of our study's initial objectives, which is why we limited our analysis to the inferior colliculus as the epileptogenic area and used the same area in matched wild-type animals as a control.

Round 2

Reviewer 2 Report

The authors have addressed most of my concerns.